# Smoothies Marketed in Spain: Are They Complying with Labeling Legislation?

**DOI:** 10.3390/nu15204426

**Published:** 2023-10-18

**Authors:** Lorena Da Silva-Mojón, Concepción Pérez-Lamela, Elena Falqué-López

**Affiliations:** 1Analytical Chemistry Area, Analytical Chemistry and Food Department, Faculty of Sciences, University of Vigo, E-32004 Ourense, Spain; lorenadasilvamojon@gmail.com (L.D.S.-M.); efalque@uvigo.es (E.F.-L.); 2Nutrition and Bromatology Area, Analytical Chemistry and Food Department, Faculty of Sciences, University of Vigo, E-32004 Ourense, Spain

**Keywords:** smoothies, nutrition label, nutritional claims, legislation compliance

## Abstract

There is no legal definition of a smoothie, so the European legislation applicable to its labeling is that of fruit juice. The smoothie market has grown in recent years, as it can include a wide variety of ingredients: fruits, fruit purees, honey, milk and vegetable milks, vegetables, herbs, cereals, cereal flours, seaweed, and crushed ice. In this study, 57 smoothies were reviewed. All of them were purchased in supermarkets and classified into eight types according to the main ingredients. Fifteen legal statements/items were reviewed on the pack labels: eleven mandatory and three optional. Moreover, nutrition labels, nutritional claims, images, marks, and other symbols were also reviewed. Only 22.8% of the samples complied with EU and Spanish labeling legislation. More incorrectness was related to the name of the food and the fruits included as main ingredients; other errors related to the allergy/intolerance statements, and some nutritional claims concerning vitamin C were also detected to a lesser extent. General advice is provided for consumers to interpret smoothie labels correctly. Lawmakers should amend legislation to accept the term “smoothie” as a legal name.

## 1. Introduction

The term “smoothie” was mentioned in the USA for the first time in the 1960s [1], and it re-emerged in the 2000s [2]. Smoothies are blended shakes that may include several ingredients of vegetal origin: fruits, vegetables (carrot, pumpkin, broccoli, spinach, cucumber), vegetable milks (from almond, oat, soya, nuts), herbs, cereals, cereal flours and algae. They may also include ingredients of animal origin: yogurt, milks and others. Milk and yoghurt provide protein content, as recommended by the USA Food and Nutrition Service [3]. In general, smoothies are a popular way of drinking fruit [4]. Some authors also recommend the addition of non-conventional ingredients such as vegetables, cereals and their derivatives to formulate innovative preparations for the market [5]. On the Japanese market, they can be considered a trendy health food product [6]. These products can be sold as fresh or packaged, and they can be considered part of a “clean eating” diet. Clean eating can be defined as choosing whole and minimally processed foods and limiting consumption of artificial or processed foods [7]. Usually, they are purchased prepared freshly from “juice bars” or as a processed (slightly pasteurized) product from the hypermarket’s refrigerated section [8] or without refrigeration. Normally, they do not contain additives/preservatives.

Several authors have pointed out the necessity of studies to help uncover and overcome barriers to sustainable eating and drinking solutions such as the consumption of smoothies [9]. Smoothies are good examples of the food industry’s reply to the growing awareness of consumers regarding healthier foods with simple ingredients and clean labels [10], without additives/preservatives, easy-to-recognize ingredients, and no artificial flavors or synthetic chemicals. Its market has been increasing in the last few years [11], as it can include a great variety of ingredients and can be consumed as a “fast food” or easy-to-eat product. So, consumers avoid preparation steps (peeling, cutting, mixing, etc.), and they are also a good choice for people with chewing difficulties (babies, elderly). In most cases, fruit and vegetable mixtures are selected based on color, flavor, texture, and above all, to ensure a high concentration of nutrients with a low energy content. Moreover, smoothies are a way to incorporate more fruits and vegetables into the diet [12], and they are also additive-free with “fresh-like” sensory properties. One study recommended the use of a smoothie to replace a meal twice a day in overweight and obese adults [13].

The word “smoothie” comes from the English term “smooth” (tender, creamy), and it defines a creamy non-alcoholic drink with a thick texture similar to that of a milkshake [14]. However, there is no legal definition of a smoothie [15]. Therefore, the labeling legislation in the European Union (EU) required for these food products to check their adequacy level should be the one related to fruit juices [16].

In the EU, the smoothie market was worth USD 3.10 billion in 2021, and it is projected to grow at a 6.8% CAGR (compound annual growth rate) to reach USD 4.30 billion by 2026. The smoothie market can be segmented into fruit-based smoothies and dairy-based smoothies, with the former being the fastest-growing segment [17]. As emerging products, different profiling studies are needed to ensure their claimed properties [10].

The label and packaging type are a product’s calling card. The current trend among food manufacturers is to add labels to food products to promote their contribution to health. In some cases, names of particular types of food (e.g., fruits) are often perceived to be healthier than their corresponding descriptions [18]. An example of this is the addition of food labeling to fruit/vegetable products based on the health contribution according to the WHO (World Health Organization) guidelines [19]. Inauthentic food products may appear as a result of adulteration and fraud through practices including, but not limited to, mislabeling or misrepresentation [20].

The food label is itself a governance space worthy of critical examination. We define the food “label” broadly, in line with legal definitions, as including all the tags, brands, marks, statements, representations, designs and descriptions on food and its packaging and made or displayed to consumers when it is sold [21]. The product label represents one way of influencing the consumer’s purchasing decision, with up to 70% being strongly influenced by the information presented on the label [22]. In particular, nutrition labeling is a tool to help people make healthy choices when choosing foods and beverages for consumption [23]. Apart from that, nutrient claims should be of concern to policymakers seeking to improve diets and tackle obesity [24].

Food regulations need to be clear and updated frequently if they are to be known and followed [25]. The labeling legislation for foods is mandatory or voluntary, depending on the concrete indication/item exposed in the label. Rodríguez-López et al. [26] reported the importance of specific regulations to guarantee consumer protection.

Consumers were reported to have two factors, background color and information on the label, as being the most important factors given on a food package [27]. Considering that the studies related to compliance with labeling legislation among smoothies are very few, the labels exposed on smoothies were studied in this work. The main objective was to expose the legal requirements and discuss the compliance degree with Spanish and EU legislation [28,29,30,31,32,33,34,35,36,37] in terms of the labels of 57 smoothies marketed in Spain. Another objective was to provide recommendations for consumers to know how to interpret a smoothie label correctly and for lawmakers to facilitate this task.

## 2. Materials and Methods

### 2.1. Samples

A total of 57 smoothies from national and international brands were bought in supermarkets around Ourense City (NW Spain). Their compositional data were photographed and compiled to classify the samples. See an example in the Appendix A. Eight types were assigned, considering main ingredients, and subsequently, coded with letters and numbers (see Table 1): 50.9% containing fruits (F); 10.5% with fruits and vegetables (FV); 7% prepared with fruits and milk (FD); 3.5% with fruits, vegetables, and milk (FVD); 10.5% including fruits and cereals (FC); 3.5% containing fruits, vegetables and cereals (FVC); 8.8% with fruits, cereals and milk (FCD) and 5.3% including fruits, vegetables, cereals and milk (FVCD).

### 2.2. Indications/Items Reviewed on the Labels

The indications/items reviewed on the labels were the ones established in the EU legal requirements for mandatory and optional food information [28] and in the Spanish legislation [29,30,35,36].

#### 2.2.1. Mandatory Food Information

The compulsory information includes the following 11 indications/items:-*The name of the food*. The name of the food should be its legal name. In the absence of a legal name, the name of the food should be its customary name, and in the absence of such a name, a descriptive name should be used. The name of the food may not be replaced by the brand name.-*The list of ingredients*. The ingredients list should be headed by the word “Ingredients”. It should list all the ingredients of the food, in descending order of weight, as incorporated at the time of their use in the manufacture of the food, except for ingredients constituting less than 2%, which may appear after the other ingredients in a different order. These can be simple or compound ingredients, or additives, as regulated by Regulation 1333/2008 [31], and/or aromas, Regulation 1334/2008 [32].-*The quantity of certain ingredients or categories of ingredients*. The quantity should be expressed as a percentage and should appear on or adjacent to the sales description. If an ingredient appears in or is associated with the sales name, it should be prominently displayed on the labeling, either in words, pictures or graphic representations, as it is essential to distinguish the food from other foodstuffs or other foods. Substances that cause allergies/intolerances must be detailed in this list and should be highlighted. If vitamins and/or minerals appear in the list of ingredients, they are considered to be added and must follow the legislation that regulates them [33].-*The net quantity of the food*. It must be expressed in units of volume (liters, centiliters or milliliters) for liquid products or in units of weight (kilograms or grams) for other products. Where a package consists of one or more individual packages containing the same quantity of the same product, the net quantity should be given by indicating the net quantity of the individual package and the total number of packages.-*The date of minimum durability or shelf life*. If the date of minimum durability includes the indication of the day, it must be indicated with “best before…”; otherwise, it should be preceded by “best before the end of…”. The above indications must be accompanied by the date itself or a reference to the place where the date is indicated on the label. The indication of the date must be clear and in the following order: month and, where appropriate, year. If the duration of the food is less than three months, it is only necessary to indicate the day and the month; if the duration is more than three months but less than 18 months, it is sufficient to indicate the month and the year.-*Special storage and/or use conditions*. They must be indicated so that the consumer can store or use the food correctly, guaranteeing its minimum durability date.-*The business’ name and address*. The food business operator responsible for the food information should be the operator under whose name or business name the food is placed on the market or, if not established in the European Union, the importer of the food into the EU market.-*The country of origin or place of provenance*. The indication of the country of origin or provenance place should be mandatory where its omission is likely to mislead the consumer or where the information accompanying the food or the label as a whole may suggest that the food has a different country of origin.-*Instructions for use*, where necessary. Instructions for the use of a food should be provided where the absence of such instructions would hinder the correct use of the food by the consumer.-*Nutritional claims*. The mandatory nutritional information must include: energy values, amounts of fat, saturated fatty acids, carbohydrates, sugars, protein, and salt. Nutritional information should be expressed per 100 g or 100 mL. Vitamins and minerals should, in addition, be expressed as a percentage of the reference intakes. Where they appear, the statement “Reference intake for an average adult (8400 kJ/2000 kcal)” should be included adjacent to them. The portion or unit used should be indicated next to the nutritional information. The nutritional value and the nutrients mentioned should appear in the same field of view and should be presented together in a clear format and the prescribed order. If there is sufficient space, this information should be presented in a table format with the data in columns; if there is insufficient space, it should be presented in a linear format. It must appear in the main field of vision and a legible font size.-*Modified atmosphere packaging*. Where the shelf life of a food has been extended by packaging gases, it must contain this indication.

#### 2.2.2. Voluntary Food Information

The optional information may or may not be regulated (see Table 2), and it may include the following food information:-*Nutritional and health claims* [34]. A “nutritional claim” is any claim that states, suggests or implies that a food has specific beneficial nutritional properties due to: the energy (calorific value) it provides, whether reduced, increased, or not provided; or the nutrients or other substances it contains in reduced, increased or not provided proportions. Their function is to make consumers perceive that these foods have nutritional, physiological or other health benefits, and thus to trigger the decision to consume these products rather than others. A “health claim” is any claim that states, suggests, or implies that a relationship exists between a food category, a food or one of its constituents and health.-*Lot* [35]. A “lot” is defined as a set of sales units of a foodstuff produced, manufactured, or packaged under virtually identical circumstances. It should be determined by the producer, manufacturer, or packer of the product, or by the first seller established in the EU. The lot should be indicated by the letter “L”, except where it is clearly distinguishable from other indications on the label. The batch indication may be omitted if the date of minimum durability or the “use by” date contains at least the day and month.-*Nominal quantity* [36]. The nominal quantity is the mass (kilogram or gram) or volume (liter, centiliter, or milliliter) of the product marked on the container labeling, i.e., the quantity of product estimated to be contained in the package. These quantities must be easily legible, visible, and indelible. Where two or more packages form a multipack, the nominal quantities also apply to each individual package. The CE marking should be placed in the same field of vision as the indication of the nominal mass or volume, and it is represented by the symbol “℮”. The ℮-mark shows that a product complies with EU rules on the indication of the volume or weight and with the measuring methods to be used by the seller of packaged products.

“Suitable for vegans” is a nutritional claim that means the smoothie is adequate for a diet composed exclusively of products of vegetal origin (fruits, cereals, legumes), algae, mushrooms, and others.

“Organic production”, when found on a smoothie label, means that the ingredients are produced in an overall system of farm management and food production that combines the best environmental and climate action practices, a high level of biodiversity, the preservation of natural resources, and the application of high animal welfare standards and high production standards, in line with the demand of a growing number of consumers for products produced using natural substances and processes [37].

## 3. Results

Product labels are very important and promote performance and emotional benefits related to nutrient formulations that go beyond conventional nutritional science [38].

The incorrectness and legal compliance of each label and the associated terms/items have been compiled in Table 3 and Table 4 as well as in Appendix A. In the latter tables, the letter “V” appears if the item is correct (when it complies with the legislation in force) and the letter “X” if it is incorrect, (i.e., if it does not comply with the legal requirements), both for the mandatory information (Appendix A) and for the nutritional and optional mentions (Appendix A).

Six mandatory indications were correctly reported: date of minimum durability, nominal quantity, storage conditions or conditions of use, country of origin or place of provenance, instructions for use, and producer/operator address. In this work, only 13 smoothies, less than a quarter of the samples (22.8%), complied with labeling legislation. These samples were: six of them containing fruits (F), one containing fruits and dairy products (FD), two with fruits and cereals (FC), and for samples containing fruits, cereals, and dairy products (FCD) (see Appendix A). The label statements tested were classified into several items, as set out in the legal requirements for mandatory food information (11 items) and optional information (3 items). The nutrition label (very important to show the nutritional content of a smoothie), the nutritional claims (voluntary on the food labels), and finally, logos, marks, and images were also reviewed.

### 3.1. Mandatory Food Information Compliance

The compliance with the legal requirements for all the mandatory terms in the smoothies studied is shown in Table 3 and Appendix A, which report the number of incorrect entries per indication/item and type of smoothie.

In terms of the mandatory food information, it was observed that the highest number of errors on labels was in the legal definition of the product (name of the food) (Figure 1). Other errors in the food name were the indication of several fruits when this was not the case, or no fruit at all (various fruits), the erroneous designation of baby food (infant food), the lack of information on concentrated juices (from concentrated) and highlighting minority ingredients in the name (highlighted ingredient).

In several samples, the name of the food was replaced by the brand name (Danonino, Nestlé, Hero), whereas the legislation recommends using the legal name of the food. The rest mentioned “fruit” or fruit names (apple, strawberry, banana, pear, red fruit) or contained images showing pictures of these fruits. The most common fruits found in the studied samples were banana, strawberry, orange and apple, whereas the most common vegetables were carrot and beetroot. Only one dairy smoothie contained oat milk; the rest were made with cow milk.

Almost half of the studied smoothies (47.4%) reported this item (food denomination) incorrectly, with the specified fruits being the indications that caused the greatest inadequacy (26.3%); some of them had fantasy denominations, for example, “FruUutitas”, “Yogurín”, or “Mi jelly”. Existing studies show that clear, simple messages are preferred by consumers [39].

The list of ingredients was correctly displayed on 91.2% of the smoothies, with the most common error being the specified amount of certain ingredients (5.3%) (see Figure 2).

There were six mandatory indications/items that were correctly shown on the labels: date of minimum durability, nominal quantity, storage conditions or conditions of use, country of origin or place of provenance, instructions for use and producer/operator address.

### 3.2. Voluntary Food Information Adequacy

The numbers of errors in the optional food information for the studied smoothies are shown in Table 4 and Appendix A. The 16 errors detected were related to the indications: one artificial aroma, 100% natural, source of fiber, with vitamin C and antioxidant messages. The lot and nominal quantity were correctly designated on all the labels.

Nutritional claims related to the addition of vitamins (C and D) and/or minerals (Ca) were displayed on the labels of 20 smoothies (35% of the samples). A total of 16 products had the claim “with vitamin C”, although half of them included this claim erroneously, as they did not reach 15% of the recommended dietary allowance of ascorbic acid.

The claim “% of fruit” was displayed in an adequate form on 26 smoothies (45.6% of the samples).

Regarding allergens, 56% of the smoothies contained the claim “without gluten”, and it was exposed in the correct way. The claim “without lactose” was correctly exposed on 10 smoothies (17.5% of samples).

Only one smoothie contained the claim “low in fat”, which was expressed in a proper way. Regarding the claim “source of fiber”, two smoothies stated it incorrectly because they do not contain more than 3 g/100 g or 1.5 g/100 kcal of the product.

Other information, such as logos, symbols, images, etc., was correctly displayed, except for one smoothie where the image of a strawberry was shown without it containing this ingredient in its list of ingredients. The use of pictures or icons appeared to enhance food consumers’ understanding compared to text-only labels.

## 4. Discussion

According to EU law, any food product sold in a Member State must carry a name identifying the food [40]. The name of a food product identifies that specific product and contains significant product information that is especially valuable to the consumer. In some cases, the mandatory name may not be the name by which consumers identify the specific product. For example, most “smoothies” are actually “juices”, and some carry dual names to inform consumers and, at the same time, comply with the law [41]. This is not the case in our study, where only 14 products (almost 25%) had the designation “smoothie” on the label.

Several authors report that smoothies are popular dietary products [42]. In our opinion, “smoothie” should be accepted as a legal food name as other terms, like “vruchtendrank” or “succo e polpa”, are quoted in Spanish juice legislation [43]. In addition, legislators should consider that the word “smoothie” is already widely used by both consumers and food processors.

Two decades ago, it was already published that information regarding health and nutrition is becoming increasingly abundant, not only in the media but also on food packaging [18], and this information can lead certain food processors to unfair practices. Nowadays, opinions on healthy nutrition are widespread and easily accessible, exposing consumers to an abundance of often conflicting perspectives from the media, health professionals, and sources on the Internet [44]. Therefore, nutritional information exposed on the smoothies’ labels was also revised and contrasted with current European legal requirements [28]. One study performed in Chile found that fruits and fruit juices account for only 6.1% of incorrectness on the nutritional label [23]. Katsouri et al. [45] found that only the third part of the cheese samples fully complied with European labeling legislation.

Regarding sugars, the WHO’s guideline suggests a maximum daily intake of 25.0 g of sugar per day for an adult with a normal body mass index [46]. Fruits and vegetables contribute significantly to the total sugar content. Smoothies contain sugars derived from their ingredients, fruits, fruit juices, cereals, and dairy products, and they need to be considered for their full nutritional proposition when compared to an added sugar product such as soft drinks. The FDA clearly distinguishes between the total and added sugars [47]. Also, the WHO makes the distinction that free sugars are different from the intrinsic sugars found in whole fresh fruits and vegetables [48]. Some authors have even developed a method to discriminate between added sugars and naturally occurring sugars in foods [49]. Other authors have reported that the term “free sugars” includes all the monosaccharides/disaccharides added to foods/beverages by the manufacturer/cook/consumer, plus the sugars naturally present in the food product. The intake of free sugars should be reduced and minimized, with a desirable goal of <5% energy intake in children and adolescents aged ≥2 to 18 years. The intake should probably be even lower in infants and toddlers <2 years [50].

Excess intake of added sugars (the sugars and syrups added to foods and beverages)—not the total sugars—is associated with an increased risk of diabetes, obesity, dental caries, and cardiovascular disease [51]. Some reviews conclude that calorie posting may persuade consumers to choose lower-calorie options [52]. So, claims related to no addition of sugar are an incentive to buy any food product. Moreover, label information about the total sugar content (mandatory item) does not make it possible to distinguish between added sugars and sugars derived from fruits/vegetables, syrups, and others. It would be necessary to use a specific analytical technique. Moreover, labels do not always specify the composition and amount of the different ingredients containing sugars. Therefore, it is not feasible to perform a calculation only considering label information to discriminate both types of sugars. The nutritional claim “no added sugars” is a claim for consumers, although this claim does not always correlate with the actual sugar content measured by analytical techniques such as high-performance liquid chromatography (HPLC).

The claim “without added sugars” appeared correctly on 36 smoothies (63% of the samples). However, this does not mean that the other samples contained less sugar. It depends more on the type of fruit and the other ingredients. For example, a smoothie containing strawberry and banana in the same proportions would logically have less sugars in the former due to the individual sugar content per type of fruit. One study conducted in Portugal found that 82% of fruits and vegetable purees and 67% of juice/smoothies/teas/drinks surpassed the level of free sugar [53]. And in two other works, both performed in the UK, the authors found that smoothies were the food product containing the most sugar: 11.5 g/100 mL [54] and 13.0 g/100 mL [55].

The other claims of no addition, “without” preservatives, colorants, palm oil, milk, powdered milk, added cream, or starch, were correctly labeled. Only one smoothie (type FCD) did not indicate that it contained artificial flavorings when it was written on its list of ingredients. This is probably because additives are perceived to be not healthy for consumers [56].

The nutritional claim “natural” appeared on the labels of three smoothies (type F) and one smoothie (type FC). Some authors have studied consumers’ interpretations regarding the food label term “natural”, with further examination and support that cognitive choices are aligned with food intake behavior [57]. Most of the consumers in that study associated the term “natural” with: “no preservatives”, “organic” and “made with real ingredients”. In one work performed in Brazil, “100% natural” received a positive interpretation by consumers, although was not well understood [58]. In another study, the color blue had the biggest impact on making the consumer believe in the health benefits of the product. This was followed by the indication of an organic origin, then the statement emphasizing the natural quality of the ingredients [59].

In one work related to the consumer response to labeling in the USA, “no added chemicals”, “no added sugar”, and “all natural” were the most important labeling terms when they purchased juice, whereas “pasteurized” ranked the lowest [60].

The upsurge in the adoption of vegan lifestyles, increased consumption of fruits and vegetables, including smoothies and juices, and the use of plant foods in nutritional or body-building supplements could aggravate allergies [61]. Regarding allergy mentions, sometimes, the undeclared allergens could reach almost half a percent of the analyzed samples in foods tested in Australia [62]. Nevertheless, in another study performed in Brazil, labels included allergen warnings, indicating that these products agreed with the legislation [63]. Improved product allergen labeling will reduce allergic reactions and simplify allergy management [64]. It is therefore important that allergens and other intolerance-causing compounds coming from dairy products (such as lactose) and cereals (such as gluten) be correctly displayed on the label.

The organic production indication and corresponding logos appeared on 21 smoothies (37% of the samples) and were correctly described. In one paper, one quarter of the products had labels pointing to sustainable aspects. However, most of these products were found in an organic food retailer and were organic juices [65]. Another study showed that explicit organic visual cues (e.g., an organic label) on packaging can affect consumer responses, like perceived healthiness [66].

Nutritional claims related to the addition of vitamins (C and D) and/or minerals (Ca) were displayed on the labels of 20 smoothies (35% of the samples). A total of 16 products had the claim “with vitamin C”, although half of them included this claim erroneously, as they did not reach 15% of the recommended dietary allowance of ascorbic acid. One study performed with processed mandarin beverages in Brazil found that six brands (46% of studied marks) were considered noncompliant with the Brazilian legislation regarding the ascorbic acid content reported on the labels [67]. Nevertheless, another study performed on baby foods found that the ascorbic acid reported on the labels complied with Brazilian legislation [63].

These results indicate the need for greater surveillance of nutritional labeling, and several authors have reported that claims can be optimized to enhance appropriate consumer understanding [39].

Because of the interest in vegetable products containing polyphenols, many beverages with the words “antioxidant” and “polyphenols” have been developed on the market [68]. Therefore, antioxidant claims may appear on packaging. Two smoothies included the claims “detox” and “antioxidant” on their labels. The main reason for displaying them is to attract the consumer’s attention; however, there is a legal gap with regard to these claims. Other authors have recently reported that claims develop freely as the responsibility of business operators with insufficient standards of performance results [69]. Thus, they cannot be considered as adequate health or nutrition claims from a legal point of view.

Other authors have reported that front-of-pack labels do not steer all consumers toward healthier choices and may even have adverse effects for some [70].

## 5. Conclusions

The main conclusion is that most of the incorrectness was related to the naming of the food. Almost half of the smoothies incorrectly described the food, considering that the word “smoothie” does not appear as a fruit derivative in the European legislation. However, the word “smoothie” defines/identifies these products better than the ones allowed by legislation, such as “fruit juice” [16,29]. So, legislators should amend the laws and consider including the word “smoothie” as a legal food name.

The nutritional claim “no added sugars” is a claim for consumers, although this claim does not always correlate with the actual sugar content measured by analytical techniques such as high-performance liquid chromatography (HPLC). The sugar contents of smoothies depend on the fruits and the other ingredients. The addition of vitamin C was also incorrectly mentioned on a high proportion of the samples that contained this nutrient in their list of ingredients. Six mandatory indications were correctly reported: date of minimum durability, nominal quantity, storage conditions or conditions of use, country of origin or place of provenance, instructions for use, and producer/operator address. Other optional mentions, for example, “organic product”, were also correctly declared.

Regarding nutritional claims, food labels could avoid a situation where claims mask the overall nutritional status of a smoothie. A general recommendation for correctly interpreting smoothie labels is to look at the nutrition label rather than at claims such as “no sugars added” or “with vitamin C”. Considering the increment in several diseases such as diabetes or obesity, or in populations with special diets (celiac, lactose intolerant, vegans, etc.), some of these claims should be enforced by legislation instead of being optional.

Policymakers should encourage research to address the issue of identifying the best items on the labels to support consumers toward healthier and more informed food choices. In this respect, FoP (front-of-package) labels seem a promising strategy [71], and over 30 countries have now implemented new labeling systems based on this concept to improve the nutritional quality of food products and to encourage producers to offer healthier foods and beverages.

Moreover, governments should survey the labeling legislation’s adequacy and penalize non-compliance.

According to our opinion, one work for the future would be the extension of this study to other countries in the EU to check if the legal compliance is better, worse or similar to that found in this study. The comparison of smoothie labels’ legal compliance with fruit juices would be also recommendable.

## Figures and Tables

**Figure 1 nutrients-15-04426-f001:**
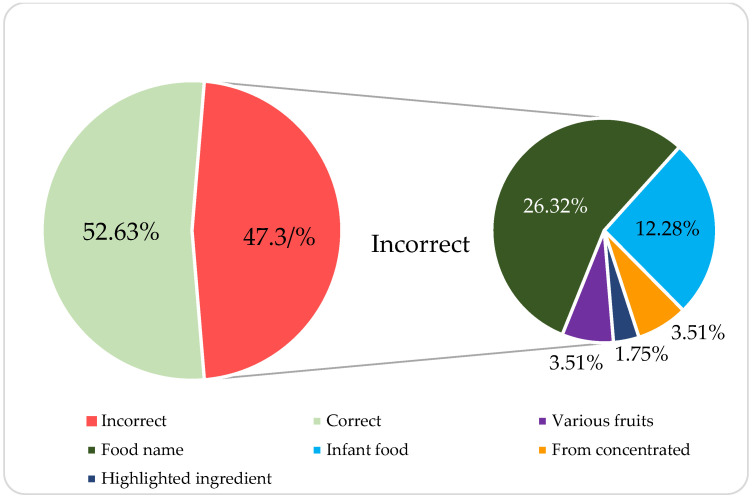
Error rates of the mandatory labeling indications.

**Figure 2 nutrients-15-04426-f002:**
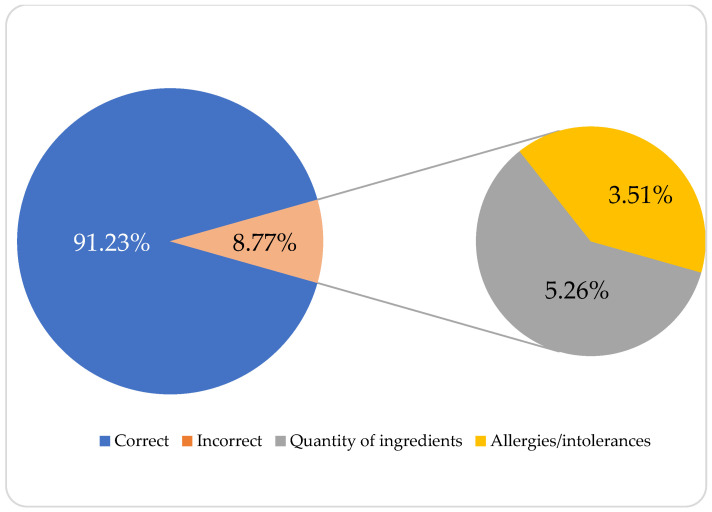
Error rates of the optional labeling indications.

**Table 1 nutrients-15-04426-t001:** Smoothies’ codes, main ingredients and number of samples per group.

Smoothie Code	Main Ingredients	Number of Samples	Group of Smoothies
F	Fruits	29	F1 to F29
FV	Fruits and vegetables	6	FV1 to FV6
FD	Fruits and dairy products	4	FD1 to FD4
FVD	Fruits, vegetables and dairy products	2	FVD1 to FVD2
FC	Fruits and cereals	6	FC1 to FC6
FVC	Fruits, vegetables and cereals	2	FVC1 to FVC2
FCD	Fruits, cereals and dairy products	5	FCD1 to FCD5
FVCD	Fruits, vegetables, cereals and dairy products	3	FVCD1 to FVCD3

**Table 2 nutrients-15-04426-t002:** Optional food information on the label that can be or not be regulated by legislation.

Regulated	Not Regulated/Other Information
Nutritional and health claims [34]	Nutritional and health claims
Lot [35]	Brand and highlighted ingredients
Nominal quantity [36]	Bar code
Suitable for vegans [28]	Recycling pictograms
Organic production [37]	Presentation images
	Business/operator contact
	Age recommended
	Color combinations

**Table 3 nutrients-15-04426-t003:** Number of errors/samples in the mandatory food information exposed on the labels of the studied smoothies.

Indication or Item		Smoothie Types and Number per Type								Total Errors
		F 29	FV 6	FD 4	FVD 2	FC 6	FVC 2	FCD 5	FVCD 3	
Food name		13/29	4/6	1/4	2/2	3/6	1/2	1/5	2/3	27/57
Ingredients list	Simple ingredients	0/29	0/6	0/4	0/2	0/3	0/2	0/5	0/3	0/54
	Additives	0/7	0/1	1/2	0/1			0/3	0/1	1/15
	Aromas	0/3	0/1					0/4	0/2	0/10
	Compound ingredients					0/3		0/1		0/4
	Quantity of certain ingredients	2/29	0/6	0/4	0/2	0/6	0/2	0/5	0/3	2/57
	Allergens/ intolerances			1/3	1/1	0/3		0/5	0/3	2/15
	Vitamins/ minerals	0/18	0/2	0/1		0/4	0/2	0/1	0/1	0/29
Net quantity		0/29	0/6	0/4	0/2	0/6	0/2	0/5	1/3	1/57
Shelf life		0/29	0/6	0/4	0/2	0/6	0/2	0/5	0/3	0/57
Conservation/use		0/29	0/6	0/4	0/2	0/6	0/2	0/5	0/2	0/56
Business name and address		0/29	0/6	0/4	0/2	0/6	0/2	0/5	0/3	0/57
Origin country *										0/57
Instructions for use		0/19	0/3	0/1	0/2	0/5		0/2	0/3	0/35
Nutritional claims		16/29	3/6	0/3	0/1	4/6	2/2	0/5	1/3	26/55
Modified atmosphere packaging		0/15	0/3		0/2	0/5		0/2	0/1	0/28
*Total error/smoothie type*		*31*	*7*	*3*	*3*	*7*	*3*	*1*	*4*	*59*

* It is not necessary to be written on the label; 0/0 is highlighted in a gray color.

**Table 4 nutrients-15-04426-t004:** Number of errors/samples in the optional food information (nutrition claims, lot and nominal quantity) for the studied smoothies.

Claim or Item	Smoothie Types and Number per Type								Total Errors
	F 29	FV 6	FD 4	FVD 2	FC 6	FVC 2	FCD 5	FVCD 3	
Without added sugars or 0% added sugars	0/21	0/4	0/1	0/1	0/5	0/0	0/1	0/3	0/36
Without preservatives	0/12	0/3	0/2	0/1	0/1				0/19
Without colorants	0/8	0/1	0/2	0/1	0/1		0/1		0/14
Without artificial aromas			0/1				1/1		1/2
100% natural	0/3		0/1	0/2			1/1		1/7
% fruit	0/18	0/3			0/1		0/1	0/3	0/26
Bio, eco	0/8	0/4	0/1	0/1	0/1	0/2	0/3	0/1	0/21
Source of vitamin C or with vitamin C	4/11	1/1		1/1	1/3	1/1			8/17
Source of fiber	1/1				1/1				2/2
Rich in calcium, source of calcium or with calcium			0/2				0/1		0/3
Without gluten	0/18	0/6	0/2	0/1	0/2	0/2	0/2	0/2	0/35
Without lactose	0/6	0/1			0/1		0/1	0/2	0/11
Without milk, powdered milk or added cream	0/3			0/1					0/4
Without palm oil		0/2			0/1		0/1	0/2	0/6
Antioxidant, detox, relaxing or energetic		1/1			1/1	2/2			4/4
Lot	0/29	0/6	0/4	0/2	0/6	0/2	0/5	0/3	0/59
Nominal quantity	0/3		0/2						0/5
*Total error/smoothie type*	*5*	*2*	*0*	*1*	*3*	*3*	*2*	*0*	*16*

0/0 is highlighted in a gray color.

## Data Availability

The data presented in this study are available on request from the corresponding author.

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
