# Peer review of "Smoothies Marketed in Spain: Are They Complying with Labeling Legislation?"

_nutrients, 2023, doi:10.3390/nu15204426_

Round 1

Reviewer 1 Report

Smoothies marketed in Spain: are they complying with label legislation?

This paper addresses the state of the market in smoothie labeling. The paper is well-conceived and covers all important segments of food  product labeling.

Abstract, title, and references         

The abstract is concise and written, with a good command of English, and a clear representation of the paper's aim. Containing 184 words, it meets the journal's demands (200 max). Furthermore, it is adequately structured: the background of the proposed research, analytical methods used, and main conclusions were mentioned.

The title of the paper adequately reflects the subject under investigation in the proposed study.

References are numbered in order of appearance in the text, as demanded by the formatting rules of the journal. Although there is no limitation in the number of references, a reference list of 66 citations is entirely sufficient to cover the topic proposed.

Introduction 

The authors represented the importance of the issue described.

Materials and methods

Line 197: Food information such as Suitable for vegans and Organic production are not described and elaborated textually  in this part.

Results

Line 212: Table 2 should be transferred to sector 3.1 of the results since the indications/items are described in detail in the material and methods.

Discussion

Line 234 and line 276: the sentence is repeated. Delete one.

Line 287: Please  provide a reference related to the results of the research in the country within the European Union.

Conclusions are supported by the described segments of the paper.

English is written in a good manner. Clear and understandable.

Author Response

Thanks for your comments.

See the answers in the attached document.

Reviewer 2 Report

Dear authors, I find the study interesting but adjustaments are needed for the manuscript to be suitable for this journal.

Introduction

lines 87 to 92 are not introduction, they are methods. Take out from this part.

Methods

Clarify to readers where is Ourense city.

you evaluated 57 smoothies. how did you choose the supermarkets? and the samples? This sample represents what is available in the market of this city?

Were they only produced in Spain or other countries?

Table 2 is already presenting results of authors´evaluation. It should be transferred. Table 4 is also results

REsults

What were the types of the 13 smothies that complied with the legislation? only made with fruits? vegetables? with milk?

Lines 221 to 233 - discussion. this is not results

Figure 2 in not necessary. It can be written in the text. It is a simple figure.

lines 260 to 263 - discussion

lines 265 to 271 - discussion

Where are the results for the nutritional evaluation?

Which were the one related to nutrient claims?

Since it is a study to evaluate smoothies in the market, it should bring more characteristics of the samples.

which are the most common fruits, and vegetables, anda milk substitutes?

Nothig was brought to discuss this

I encourage to bring a table with the mean energy value, carbohydrates, fibers, vitamin C to show their nutritional contribution.

Discussion

line 276 to 279 - not discussion

lines 288 to 301 it is a good discussion, but these results are not shown in the results section. It is very important to show these results.

again in lines 307 to 315 - discussion not related to data presented.

lines 348 to 350 - results

Conclusion

Rewrite because most of it is not conclusions of the results of the study, just the first paragraph.

What sre the limitations?

Author Response

Thanks for the comments.

See the answers in the attached document.

Reviewer 3 Report

The topic of the study is important as well as really interesting. Moreover, the topic is quite original and the obtained results in general can be the fruitful basis for researches referring to provision of food information to consumers.

Generally paper is well organized. However, I have included some main remarks:

(1)   You put the same sentence on the page no 7 (line 234) and page no 9 (line 276). I think you do not have to repeat this aspect in the same way.

(2)   In the Methodology section please indicate some details regarding the product choice (Did you choose the all products from this particular supermarket? or maybe Did you choose some of them from the accessible offer?).

(3)   In the abstract you mentioned that: ‘General advice is given to consumers to interpret smoothie labels correctly.’

Could you please explain/add the ways/methods that are the best alternatives to manage on this interpretation. Sometimes the main challenge is to encourage the consumer to read the info on the label and the process of understanding the label is the second crucial step, etc. What is your opinion on this matter?

Minor editing of English language required.

Author Response

(The authors gave the same response as above.)

Round 2

Reviewer 2 Report

Authors did not mention the limitations of their study at the end of the discussion section or in the conclusion section. This is important.

ok

Author Response

Dear referee, thanks for your comments. English has been reviewed by using Grammarly programm.

The limitations of the study have been added at the end of Conclusions.

Reviewer 3 Report

Dear Authors,

Thank you for including the corrections.

Kind regards,

Minor editing of English language required.

Author Response

Dear referee, thanks for your comments. English has been reviewed by using Grammarly programme.